# Generalized and Discriminative Few-Shot Object Detection via SVD-Dictionary Enhancement

**Aming Wu**[1]    **Suqi Zhao**[1]    **Cheng Deng**[1]*    **Wei Liu**[2]

[1] School of Electronic Engineering, Xidian University, Xi'an, China
[2] Tencent Data Platform

amwu@xidian.edu.cn, sqzhao@stu.xidian.edu.cn, chdeng@mail.xidian.edu.cn, wl2223@columbia.edu

## Abstract

Few-shot object detection (FSOD) aims to detect new objects based on few annotated samples. To alleviate the impact of few samples, enhancing the generalization and discrimination abilities of detectors on new objects plays an important role. In this paper, we explore employing Singular Value Decomposition (SVD) to boost both the generalization and discrimination abilities. In specific, we propose a novel method, namely, SVD-Dictionary enhancement, to build two separated spaces based on the sorted singular values. Concretely, the eigenvectors corresponding to larger singular values are used to build the generalization space in which localization is performed, as these eigenvectors generally suppress certain variations (e.g., the variation of styles) and contain intrinsical characteristics of objects. Meanwhile, since the eigenvectors corresponding to relatively smaller singular values may contain richer category-related information, we can utilize them to build the discrimination space in which classification is performed. Dictionary learning is further leveraged to capture high-level discriminative information from the discrimination space, which is beneficial for improving detection accuracy. In the experiments, we separately verify the effectiveness of our method on PASCAL VOC and COCO benchmarks. Particularly, for the 2-shot case in VOC split1, our method significantly outperforms the baseline by 6.2%. Moreover, visualization analysis shows that our method is instrumental in doing FSOD.

## 1 Introduction

With the rejuvenation of deep neural networks, for object detection, many progresses [11, 12, 1, 26, 22] have been achieved. Though these methods obtain outstanding detection performances, they usually require a large number of labeled samples for training, which are labored yet expensive to collect and annotate. On the contrary, human beings are born with the ability to learn a new visual concept with only few samples. To imitate such an ability of human beings, the task of few-shot object detection (FSOD) [2, 17, 36] has been proposed, which aims to improve the detection performance for new objects that contain few annotated training samples.

The main challenge of FSOD lies in how to learn generalized and discriminative object features from both abundant samples in base object categories and few samples in new object categories, which can improve the representation ability of object features and alleviate overfitting on new objects. Following the popular methods for few-shot image classification, earlier attempts [38, 37, 33, 8] in FSOD utilize the meta-learning strategy [29, 31, 10], whose goal is to learn detectors across tasks and then transfer to the few-shot detection task. However, compared with traditional two-stage fine-tuning based approaches [34, 35, 30], the meta-learning strategy fails to effectively improve generalization

---

*Corresponding Author.

35th Conference on Neural Information Processing Systems (NeurIPS 2021).

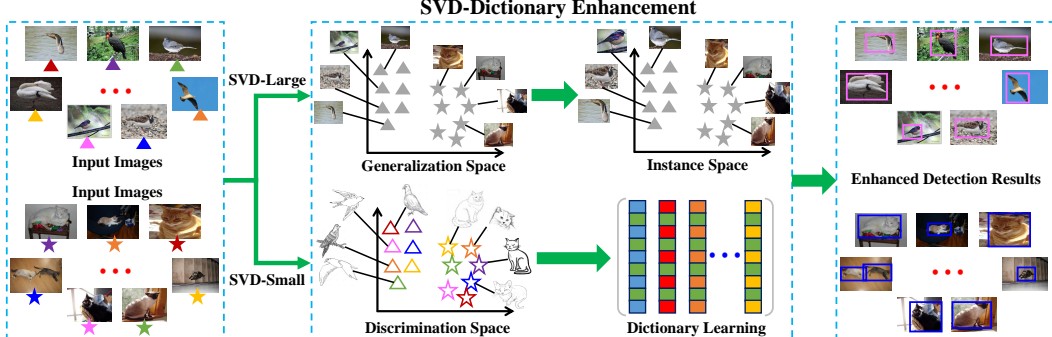

Figure 1: SVD-Dictionary enhancement for FSOD. 'SVD-Large' indicates that we use the eigenvectors corresponding to larger singular values to build the generalization space in which localization is performed. 'SVD-Small' indicates that we use the eigenvectors corresponding to smaller singular values to build the discrimination space in which classification is performed. Meanwhile, dictionary learning [41] is used to capture high-level discriminative information from the discrimination space, which is beneficial for improving the detection accuracy.

and discrimination of object features and leads to weak performance. The reason may be that during each training episode, meta-learning methods focus on transferability across different tasks and ignore learning of generalized and discriminative feature representations.

For FSOD, the generalized representations may contain intrinsical characteristics of object features, which is beneficial for adapting knowledge from base object categories to new object categories. Meanwhile, the discriminative representations may contain certain category-related information, which is helpful for boosting the detection accuracy. Furthermore, recent research [3] has shown that from a spectral analysis perspective, the feature representations can be decomposed into eigenvectors with importance quantified by the corresponding singular values. The eigenvectors corresponding to larger singular values contribute to the generalization ability, as these eigenvectors could suppress certain variations (e.g., the variations of style and texture). Meanwhile, since the eigenvectors corresponding to relatively smaller singular values contain richer category-related information (e.g., the structures of objects), these eigenvectors are beneficial for discrimination. Therefore, in this paper, we explore employing Singular Value Decomposition (SVD) (as shown in Fig. 1) to promote detectors to learn generalized and discriminative object features.

Particularly, we propose a method named as SVD-Dictionary enhancement for FSOD. Given an input image, a backbone network is first used to extract the corresponding feature map. Then, SVD is performed on the feature map. Here, we select the eigenvectors corresponding to the first $k$ largest singular values to compute a generalization map. And the generalization ability is enhanced by a residual operation between the generalization map and the original feature map. Next, the residual eigenvectors are used to calculate a discrimination map. Meanwhile, to further enhance discrimination, we define a codebook containing multiple codewords and employ dictionary learning [41] to capture high-level discriminative information from the discrimination map, which is good for accurate detection. Compared with most methods [35, 33, 17] for FSOD, our method includes two virtues. One is that during enhancing generalization, our method does not introduce extra parameters. The other is that with the help of the discrimination map and dictionary learning, our method could capture high-level discriminative information of different categories, which is conductive to reducing the data-scarce impact on new object categories. During training, we first train the model on the data-abundant base object categories. Then, the model is fine-tuned on a reconstructed training set that contains a small number of balanced training samples from both base and new object categories. Extensive experiments on two benchmarks demonstrate the superiorities of our method.

The contributions of our work are summarized as follows:

- To boost both the generalization and discrimination abilities, we propose to build the generalization and discrimination spaces based on the sorted singular values.
- To further enhance the discrimination ability, we explore dictionary learning to capture high-level discriminative information from the discrimination map.
- By plugging our method into two two-stage methods, i.e., MPSR [35] and FSCE [30], our method significantly improves their performances on PASCAL VOC [6, 7] and COCO [20].

## 2    Related Work

**Few-shot image classification.** The goal of few-shot image classification [29, 24] is to recognize new categories with very few labeled samples. Recently, many progresses [5, 32, 42, 40, 13] have been achieved. Particularly, meta-learning [10] is a widely used method to solve few-shot classification, which aims to leverage task-level meta knowledge to help models adapt to new tasks with few labeled samples. Based on the meta-learning policy, Snell et al. [29] proposed a prototypical network to learn a metric space in which classification can be performed by computing distances to the prototype representation of each category. However, the performance of this method relies on the quality of the learned prototypes. When the training data is scarce, the learned prototypes could not represent the information of each category sufficiently, which affects the classification performance. Liu et al. [23] proposed a method of prototype rectification, which considers the intra-class bias and the cross-class bias and improves the performance significantly. Apart from these methods, more methods, e.g., sample synthesis and augmentation, in few-shot learning can be seen in the work [24]. Whereas, these classification methods could not be directly applied to detection that requires localizing and recognizing objects simultaneously.

**Few-shot object detection.** Towards FSOD, most existing methods [18, 8, 25, 2, 39] employ a meta-learning or fine-tuning based mechanism. Particularly, Wang et al. [33] proposed a meta-learning framework to leverage meta-level knowledge from base object categories to facilitate the generation of a detector for new object categories. Based on this work [33], Kang et al. [17] further proposed a one-stage detection architecture that contains a meta feature learner and a reweighting module. In order to alleviate the impact of complex background and multiple objects on one image, Yan et al. [38] extended Faster R-CNN [27] and Mask R-CNN [16] by proposing meta-learning over RoI (Region-of-Interest) features. Recently, the two-stage fine-tuning based approach (TFA) [34] reveals a potential for addressing FSOD. By simply fine-tuning the box classifier and regressor, this method outperforms many meta-learning based methods. Wu et al. [35] considered the impact of the scale bias on the fine-tuning process, which further improves the detection performance.

Different from the above methods, in this paper, we explore enhancing both generalization and discrimination for FSOD. And we propose a method of SVD-Dictionary enhancement that combines SVD with dictionary learning. Experimental results and visualization analysis demonstrate the superiorities of the proposed method.

## 3    SVD-Dictionary Enhancement for FSOD

In this paper, we follow the same settings introduced in Kang et al. [17]. Concretely, there are a set of base object categories that contain abundant annotated samples and a set of new object categories that contain only few (usually less than 30) annotated samples per category. The main purpose is to improve the detection performance of new object categories.

### 3.1    SVD Enhancement

For FSOD, generalization and discrimination are two important criteria that characterize the goodness of feature representation. Particularly, enhancing generalization is beneficial for adapting the knowledge learned from base object categories to new object categories, which alleviates the data-scarce impact on new object categories. Meanwhile, discrimination refers to the ability to separate different categories based on the learned representations. And enhancing discrimination is helpful for reducing the overfitting risk on new object categories, which improves the detection accuracy. To this end, we explore SVD to enhance both the generalization and discrimination abilities of detectors.

Concretely, as shown in Fig. 2, we adopt a widely used two-stage object detector, i.e., Faster R-CNN [27], as the basic detection model. Given an input image, we first employ the feature extractor, e.g., ResNet [15], to extract the corresponding feature map $F \in \mathbb{R}^{m \times w \times h}$, where $m$, $w$, and $h$ separately denote the number of channels, width, and height. Then, $F$ is reshaped as $\boldsymbol{F} \in \mathbb{R}^{m \times n}$, where $n = w \times h$. SVD is used to factorize the matrix $\boldsymbol{F}$, i.e., $\boldsymbol{F} = \boldsymbol{U}\boldsymbol{\Sigma}\boldsymbol{V}^T \in \mathbb{R}^{m \times n}$, into the product of three matrices, where $\boldsymbol{U} \in \mathbb{R}^{m \times m}$ and $\boldsymbol{V} \in \mathbb{R}^{n \times n}$ are orthogonal, and $\boldsymbol{\Sigma}$ contains the sorted singular values along its main diagonal [4]. Since the eigenvectors corresponding to larger singular values contain more information of the original matrix $\boldsymbol{F}$, we select the eigenvectors corresponding to the first $k$ largest singular values to compute the generalization map $\boldsymbol{G} \in \mathbb{R}^{m \times n}$. Next, by feat of the

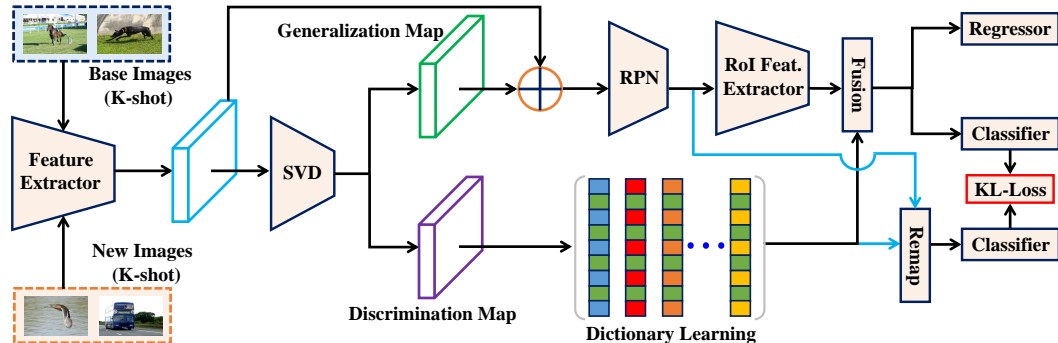

Figure 2: The architecture of generalized and discriminative FSOD via SVD-Dictionary enhancement. Here, '⊕' indicates the residual operation. 'RPN' denotes Region-Proposal Network with RoI Pooling. After extracting the corresponding feature maps of input images, SVD is utilized to compute all singular values and eigenvectors. Then, the eigenvectors corresponding to larger singular values are used to compute the generalization map. And the eigenvectors corresponding to smaller singular values are used to calculate the discrimination map. Finally, dictionary learning is used to further capture high-level discriminative information, which helps improve the ability of accurate detection.

residual operation between $G$ and $F$, the generalization ability of the extracted features is enhanced. The processes are shown as follows:

$$G = U_{m \times k} \Sigma_{k \times k} V_{k \times n}^T, \qquad E = G + F, \tag{1}$$

where $U_{m \times k}$ and $V_{k \times n}^T$ indicate that we select the first $k$ columns and rows from the matrix $U$ and $V^T$, respectively. $\Sigma_{k \times k}$ is a diagonal matrix with the dimension $k \times k$. $E \in \mathbb{R}^{m \times n}$ is the enhanced matrix. Finally, $E$ is reshaped as $E \in \mathbb{R}^{m \times w \times h}$ that is used to perform the following RPN operation. It is worth noting that in the process of enhancing generalization, we only perform the SVD operation and do not introduce extra parameters. Besides, we utilize the residual operation to obtain the output $E$, which strengthens the generalization ability and retains the discriminative information in the output. In the experiment, we observe that utilizing the operation of enhancing generalization improves the detection performance effectively.

Next, the remaining eigenvectors and corresponding singular values are used to calculate the discrimination map $D \in \mathbb{R}^{m \times n}$. The processes are the same as computing $G$. Since the map $D$ contains more category-related information [3], e.g., the structures of objects, it is helpful for enhancing the discrimination ability. Similarly, for this process, we do not introduce extra parameters, either.

### 3.2 SVD-based Dictionary Learning

**Dictionary Learning.** Based on the map $D$, we explore employing dictionary learning [41, 14] to capture high-level discriminative information, which is beneficial for strengthening the discrimination ability of detectors. Concretely, we define a learned codebook $C = \{c_j \in \mathbb{R}^m, j = 1, \cdots, Q\}$ that contains $Q$ codewords. Each element $d_i \in \mathbb{R}^m$ of the map $D$ can be assigned with a weight $a_{ij}$ to each codeword $c_j$ and the corresponding residual vector is denoted by $r_{ij} = d_i - c_j$, where $i = 1, 2, \cdots, n$. Thus, dictionary learning can be calculated as follows:

$$x_j = \sum_{i=1}^n a_{ij} r_{ij}, \qquad a_{ij} = \frac{\exp(-s_j ||r_{ij}||^2)}{\sum_{j=1}^Q \exp(-s_j ||r_{ij}||^2)}, \tag{2}$$

where $s_j$ indicates the learnable smoothing factor for the corresponding codeword $c_j$. Finally, the output of dictionary learning is a fixed length representation $X = \{x_j \in \mathbb{R}^m, j = 1, \cdots, Q\}$. Next, we take $E$ as the input of the RPN module to obtain a set of object proposals $P \in \mathbb{R}^{z \times m \times o \times o}$, where $z$ and $o$ separately denote the number of proposals and their spatial size. And the fusion result of $P$ and $X$ is taken as the input of the classifier.

$$P = \text{RPN}(E), \qquad y = \text{cls}([\phi(P), w_c X + b_c]), \tag{3}$$

where $\phi$ consists of two fully-connected layers. $w_c$ and $b_c$ are learnable parameters. '[,]' indicates the fusion operation. Here, we use the concatenation operation. 'cls' denotes the classifier. By the

constraint of the classification loss, we can promote the learned representation $X$ and codebook $C$ to absorb category-related information, which is good for enhancing detection accuracy.

**Dictionary-based Remap.** To further facilitate the learned codebook $C$ to retain more category-related characteristics, we try to remap $P$ to the dictionary space and perform classification. Concretely, each element $p \in \mathbb{R}^m$ of $P$ is remapped as a combination of codewords in the codebook $C$. The processes are shown as follows:

$$rep = \sum_{j=1}^{Q} \frac{\exp(\psi(p)c_j^T)}{\sum_{j=1}^{Q} \exp(\psi(p)c_j^T)} c_j, \tag{4}$$

where $\psi$ is a fully-connected layer that maps $p$ to the dictionary space. $rep \in \mathbb{R}^m$ indicates one element of the remapping output $Rep \in \mathbb{R}^{z \times m \times o \times o}$. Next, $Rep$ is taken as the input of the classifier to output the probability:

$$y_{rep} = \mathrm{cls}([\phi(P), \phi(Rep)]), \tag{5}$$

where $y_{rep}$ indicates the output probability. Eq. (3) and Eq. (5) share the same classifier. Finally, the KL-Divergence loss $\mathcal{L}_{\mathrm{kl}}$ is leveraged to enforce the prediction consistency between $y_{rep}$ and $y$. By performing classification in the dictionary space, the codebook $C$ could be directly facilitated to learn category-related characteristics, which is conductive to the improvement of the discrimination ability.

### 3.3 Two-Stage Fine-Tuning Mechanism

In this paper, we employ the commonly used detection loss [27] to optimize the model. Concretely, the joint training loss is defined as follows:

$$\mathcal{L} = \mathcal{L}_{\mathrm{cls}} + \mathcal{L}_{\mathrm{loc}} + \mathcal{L}_{\mathrm{rpn}} + \lambda \mathcal{L}_{\mathrm{kl}}, \tag{6}$$

where $\mathcal{L}_{\mathrm{cls}}$ and $\mathcal{L}_{\mathrm{loc}}$ separately indicate the classification and bounding-box regression losses. $\mathcal{L}_{\mathrm{rpn}}$ is the RPN loss that is used to distinguish foreground from background and refine bounding-box anchors. The hyper-parameter $\lambda$ is set to 1.0 in the experiment.

During training, we employ the two-stage fine-tuning mechanism to optimize the proposed method. Currently, there exist two fine-tuning training strategies. One is that during the base training and fine-tuning stage, all the parameters of the detector are optimized simultaneously [35]. The other is that during the fine-tuning stage, some important parameters of the detector are optimized. And the remaining parameters are fixed [34, 30]. To demonstrate the effectiveness of the proposed method, we separately utilize these two strategies to optimize the detector. Specifically, in the base training stage, we employ the joint loss $\mathcal{L}$ to optimize the entire model based on the data-abundant base object categories. During fine-tuning, the last fully-connected layer (for classification) of the detection head is replaced. The new classifier is randomly initialized. For the first strategy, we follow MPSR [35] to optimize all the parameters of the model based on a balanced training set consisting of both the few base and new object categories. For the second strategy, we follow FSCE [30] to jointly fine-tune the FPN [21] pathway and RPN while fixing the backbone.

### 3.4 Further Discussion

In this section, we further discuss SVD and dictionary learning for few-shot object detection.

For FSOD, the two-stage fine-tuning mechanism can be regarded as a method that adapts the knowledge from base object categories to new object categories, which is effective to alleviate the data-scarce impact. Most existing methods [34, 30] focus on designing an effective optimizing strategy and pay little attention to improving both the generalization and discrimination during the fine-tuning stage. Recently, FSCE [30] brings contrastive learning [19] into FSOD, which is beneficial for enhancing discrimination. However, the contrastive loss is calculated based on object proposals, which neglects the impact of generalization on object localization.

For FSOD, we propose an SVD-Dictionary method to enhance both generalization and discrimination. Particularly, the eigenvectors corresponding to larger singular values are directly used to enhance generalization without introducing extra parameters. Meanwhile, we employ dictionary learning to capture high-level discriminative information, which leads to accurate detection. Experimental results and visualization analysis demonstrate the superiorities of our method.

# 4 Experiments

In the experiments, the proposed method is evaluated on PASCAL VOC [6, 7] and COCO [20] benchmarks. We strictly follow the consistent few-shot detection data construction and evaluation protocol [17, 35, 37, 34] to ensure fair and direct comparison. Meanwhile, since our method is trained based on the two-stage fine-tuning mechanism, we take two-stage methods, i.e., TFA [34], MPSR [35], and FSCE [30], as the compared baselines. Code will be available in `https://github.com/AmingWu/SVD-Dictionary-Enhancement`.

## 4.1 Implementation Details and Few-Shot Detection Benchmarks

**Implementation Details.** For the detection model, we use Faster R-CNN [27] with the RoI Align [16] layer. The backbone is ResNet-101 [15]. The parameters are pre-trained on ImageNet [28] for initialization. In Eq. (1), we select the first $k$ largest singular values to compute the generalization map. Here, $k$ is set to half of the total number of singular values. For dictionary learning, the number of codewords is set to 24. All newly introduced parameters are initialized randomly. All the experiments are trained using the standard SGD optimizer with a momentum of 0.9 and a weight decay of 0.0001. During inference, we take the output $y$ of Eq. (3) as the classification result.

**FSOD Benchmarks.** For PASCAL VOC, the overall 20 categories are divided into 15 base object categories and 5 new object categories. All base object category data from PASCAL VOC 07+12 trainval sets is available. For each new object category, there exist **K** instances available and **K** is set to 1, 2, 3, 5, and 10. Following existing methods [17, 34, 35], we utilize the same three random partitions of base and new object categories, referred to as New Split 1, 2, and 3. And for the predictions on PASCAL VOC 2007 test set, we separately report the results of nAP50 and nAP75.

For the 80 categories in COCO, 20 categories overlapped with PASCAL VOC are taken as new object categories. The remaining 60 categories are used as base object categories. The **K** $= 10$ and 30 shots detection performance is evaluated on 5,000 images from COCO 2014 validation set.

## 4.2 Performance Analysis of Few-Shot Detection

**PASCAL VOC Results.** Table 1 shows the results on three PASCAL VOC New Splits. We can see that as the number of object instances increases, the performance continually improves significantly. This shows that few samples affect the performance of object detection. Besides, compared with the two-stage fine-tuning training mechanism, the training process of the meta-learning mechanism is more complex. However, for FSOD, meta-learning based methods [38, 33, 37] fail to obtain superior performance. The reason may be that these methods focus on learning task-level transferability and ignore the learning of feature generalization and discrimination. Next, we can see that plugging our method into MPSR [35] and FSCE [30] improves their performances significantly. Particularly, based on nAP50 and nAP75, the performance of FSCE is significantly improved. These analyses demonstrate that the proposed method is helpful for enhancing the generalization and discrimination abilities of detectors, which is beneficial for FSOD.

In Fig. 3, we show some detection examples. We can see that compared with MPSR and FSCE, our method localizes and recognizes the objects in these images accurately. Particularly, there exist three types of error detections, i.e., missing detection that misses the detection of certain objects (e.g., the fifth example in the first row), uncertain detection that classifies objects into multiple different categories (e.g., the second example in the first row), and mis-classifications of objects (e.g., the first example in the first row). For these examples, our method reduces the appearance of these errors, which shows improving generalization and discrimination is beneficial for accurate detection.

**COCO Results.** Table 2 shows the COCO results. We can also see that plugging our method into MPSR and FSCE leads to performance improvement. Particularly, for MPSR, based on the 30-shot case, plugging our method separately improves its performance by 2.4 % (AP), 2.4 % (AP75), and 3.7 % ($AP_L$). For FSCE, plugging our method is beneficial for boosting the detection performance. This further shows the effectiveness of our method. Besides, FSOD-VE [37] is a recently proposed meta-learning method, which leverages viewpoint estimation to solve FSOD. Though FSOD-VE's performance outperforms fine-tuning based methods [35, 30], the training process of meta-learning is much more complex. And the performance on small objects is weaker. This shows that improving the generalization and discrimination during the fine-tuning process is an effective solution for FSOD.

Table 1: Few-shot detection performance (%) on PASCAL VOC New Split sets. 'MPSR + Ours' and 'FSCE + Ours' separately indicate that we plug our method into MPSR [35] and FSCE [30]. 'ft' denotes fine-tuning. '†' represents meta-learning based methods. '⋆' indicates that we directly run the released code to obtain the results. The evaluation of the last two rows is based on nAP75. The evaluation of the other rows is based on nAP50.

| Method (nAP50) / Shot | New Split 1 | | | | | New Split 2 | | | | | New Split 3 | | | | |
|---|---|---|---|---|---|---|---|---|---|---|---|---|---|---|---|
| | 1 | 2 | 3 | 5 | 10 | 1 | 2 | 3 | 5 | 10 | 1 | 2 | 3 | 5 | 10 |
| FRCN-ft [33] | 13.8 | 19.6 | 32.8 | 41.5 | 45.6 | 7.9 | 15.3 | 26.2 | 31.6 | 39.1 | 9.8 | 11.3 | 19.1 | 35.0 | 45.1 |
| FRCN+FPN-ft [34] | 8.2 | 20.3 | 29.0 | 40.1 | 45.5 | 13.4 | 20.6 | 28.6 | 32.4 | 38.8 | 19.6 | 20.8 | 28.7 | 42.2 | 42.1 |
| †Meta R-CNN [38] | 19.9 | 25.5 | 35.0 | 45.7 | 51.5 | 10.4 | 19.4 | 29.6 | 34.8 | 45.4 | 14.3 | 18.2 | 27.5 | 41.2 | 48.1 |
| †MetaDet [33] | 18.9 | 20.6 | 30.2 | 36.8 | 49.6 | 21.8 | 23.1 | 27.8 | 31.7 | 43.0 | 20.6 | 23.9 | 29.4 | 43.9 | 44.1 |
| †FSOD-VE [37] | 24.2 | 35.3 | 42.2 | 49.1 | 57.4 | 21.6 | 24.6 | 31.9 | 37.0 | 45.7 | 21.2 | 30.0 | 37.2 | 43.8 | 49.6 |
| TFA w/fc [34] | 36.8 | 29.1 | 43.6 | 55.7 | 57.0 | 18.2 | 29.0 | 33.4 | 35.5 | 39.0 | 27.7 | 33.6 | 42.5 | 48.7 | 50.2 |
| TFA w/cos [34] | 39.8 | 36.1 | 44.7 | 55.7 | 56.0 | 23.5 | 26.9 | 34.1 | 35.1 | 39.1 | 30.8 | 34.8 | 42.8 | 49.5 | 49.8 |
| Retentive R-CNN [9] | 42.4 | 45.8 | 45.9 | 53.7 | 56.1 | 21.7 | 27.8 | 35.2 | 37.0 | 40.3 | 30.2 | 37.6 | 43.0 | 49.7 | 50.1 |
| MPSR⋆ [35] | 40.7 | 41.2 | 48.9 | 53.6 | 60.3 | 24.4 | 29.3 | 39.2 | 39.9 | 47.8 | 32.9 | 34.4 | 42.3 | 48.0 | 49.2 |
| MPSR + Ours | 41.5 | **47.4** | **51.5** | 57.7 | 61.2 | **29.4** | 29.6 | 39.8 | 41.2 | **51.5** | 36.0 | 39.4 | 45.4 | 50.4 | 51.3 |
| FSCE⋆ [30] | 44.2 | 43.2 | 45.7 | 58.3 | 61.0 | 25.4 | 29.5 | 42.1 | 43.6 | 48.7 | 37.2 | 43.5 | 45.8 | 53.3 | 55.8 |
| FSCE + Ours | **46.1** | 43.5 | 48.9 | **60.0** | **61.7** | 25.6 | **29.9** | **44.8** | **47.5** | 48.2 | **39.5** | **45.4** | **48.9** | **53.9** | **56.9** |
| FSCE⋆ (nAP75) [30] | 21.9 | 21.2 | 20.1 | 32.7 | 38.8 | 6.9 | 8.4 | 14.7 | 20.3 | **25.9** | 16.3 | 18.3 | 18.9 | 25.4 | 29.6 |
| FSCE + Ours (nAP75) | **25.1** | **21.4** | **25.1** | **36.5** | **39.8** | **9.4** | **11.3** | **18.5** | **24.1** | 25.6 | **18.4** | **20.5** | **24.2** | **26.8** | **30.5** |

Table 2: Few-shot detection evaluation results (%) on COCO. Here, $AP_S$, $AP_M$, and $AP_L$ separately indicate the detection performances of the small, medium, and large objects.

| Shots | Method | AP | AP75 | $AP_S$ | $AP_M$ | $AP_L$ |
|---|---|---|---|---|---|---|
| 10 | †Meta R-CNN [38] | 8.7 | 6.6 | 2.3 | 7.7 | 14.0 |
| | †MetaDet [33] | 7.1 | 6.1 | 1.0 | 4.1 | 12.2 |
| | †FSOD-VE [37] | **12.5** | 9.8 | 2.5 | **13.8** | **19.9** |
| | TFA w/fc [34] | 10.0 | 9.2 | – | – | – |
| | TFA w/cos [34] | 10.0 | 9.3 | – | – | – |
| | MPSR⋆ [35] | 9.5 | 9.5 | 3.3 | 8.2 | 15.9 |
| | MPSR + Ours | 11.0 | **10.6** | **4.4** | 11.5 | 17.1 |
| | FSCE⋆ [30] | 11.3 | 9.6 | 3.7 | 10.7 | 18.6 |
| | FSCE + Ours | 12.0 | 10.4 | 4.2 | 12.1 | 18.9 |
| 30 | †Meta R-CNN [38] | 12.4 | 10.8 | 2.8 | 11.6 | 19.0 |
| | †MetaDet [33] | 11.3 | 8.1 | 1.1 | 6.2 | 17.3 |
| | †FSOD-VE [37] | 14.7 | 12.2 | 3.2 | 15.2 | 23.8 |
| | TFA w/fc [34] | 13.4 | 13.2 | – | – | – |
| | TFA w/cos [34] | 13.7 | 13.4 | – | – | – |
| | MPSR⋆ [35] | 13.8 | 13.5 | 4.0 | 12.9 | 22.9 |
| | MPSR + Ours | **16.2** | **15.9** | 4.6 | 14.6 | **26.6** |
| | FSCE⋆ [30] | 15.4 | 14.2 | 5.5 | 14.9 | 24.4 |
| | FSCE + Ours | 16.0 | 15.3 | **6.0** | **16.8** | 24.9 |

## 4.3 Ablation Analysis

In this section, ablation analysis is performed based on the New Split 1 of PASCAL VOC. And we plug our method into MPSR to make the ablation analysis.

**Analysis of Hyper-parameter $k$.** In Eq. (1), we select the first $k$ columns and rows from $U$ and $V^T$ that correspond to the first $k$ largest singular values to build the generalization map. To enhance the generalization ability, the generalization map is expected to contain much information reflecting intrinsical characteristics of objects. In Table 3, we analyze the impact of $k$. Here, we only change the setting of $k$. The other modules are kept unchanged. We can see that different

Table 3: The performance (%) of using a different number of singular values. Here, 'proportion' indicates the percentage of the total number of singular values.

| proportion/shot | 1 | 2 | 3 | 5 | 10 |
|---|---|---|---|---|---|
| 10% | 40.6 | 43.9 | 49.1 | 55.6 | **62.1** |
| 25% | 38.3 | 44.7 | 49.4 | 56.2 | 61.7 |
| 50% | **41.5** | **47.4** | **51.5** | **57.7** | 61.2 |
| 75% | 36.3 | 42.9 | 48.7 | 55.1 | 60.9 |
| 90% | 37.9 | 41.8 | 48.1 | 55.8 | 60.2 |

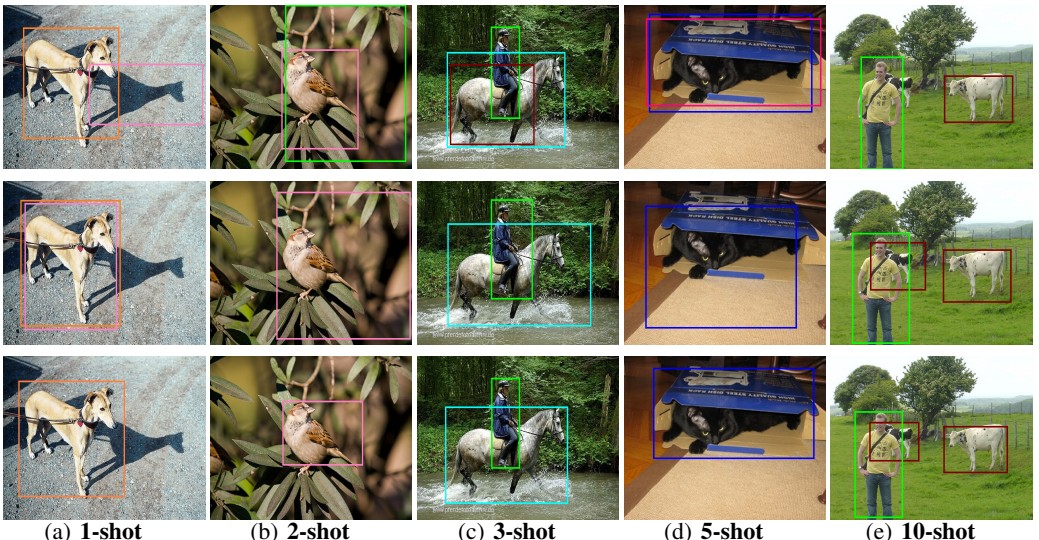

| (a) **1-shot** | (b) **2-shot** | (c) **3-shot** | (d) **5-shot** | (e) **10-shot** |

Figure 3: Detection examples based on different shots. The first, second, and third rows separately indicate detections based on MPSR [35], FSCE [30], and our method. We can see our method accurately detect 'dog', 'bird', 'person', 'horse', 'cat', and 'cow'.

settings of $k$ affect the performance of FSOD. Particularly, while $k$ is set to a large value or a small value, the performance decreases. The reason may be that using a large value of proportion introduces much information that is not related to intrinsical characteristics, which weakens the generalization ability. Meanwhile, using a small value of proportion may lead to the loss of certain object-related information, which weakens the feature representation. We observe that the performance of using 50% proportion is the best.

**Analysis of SVD-based Generalization and Dictionary Learning.** To demonstrate the effectiveness of the proposed method, we remove the module of dictionary learning and only keep the SVD-based generalization. From the 1-shot to 10-shot case, the performance is 41.2%, 44.3%, 49.7%, 54.8%, and 60.9%. We can see that employing dictionary learning is helpful for improving detection performance. Particularly, taking the 2-shot case as an example, the performance is improved by 3.1%. This indicates based on the output of SVD operation, dictionary learning is able to leverage multiple codewords to capture high-level discrimina-

Table 4: The performance (%) of using a different number of codewords in the codebook of dictionary learning.

| number/shot | 1 | 2 | 3 | 5 | 10 |
|---|---|---|---|---|---|
| 16 | 39.2 | **48.3** | 51.3 | 55.8 | 60.5 |
| 20 | 40.1 | 48.1 | 49.6 | 56.1 | 60.7 |
| 24 | 41.5 | 47.4 | **51.5** | **57.7** | **61.2** |
| 28 | 41.9 | 47.6 | 50.9 | 56.5 | 60.4 |
| 32 | **42.1** | 46.2 | 51.3 | 56.9 | 60.3 |

tive information that is helpful for accurate detection. Meanwhile, this also shows that the learned codewords contain category-related information, which enhances the discrimination ability of the detector. Besides, we can see the current performance of only using SVD-based generalization still outperforms MPSR. Taking the 2-shot and 5-shot cases as examples, the performance is separately improved by 3.1% and 1.2%. This indicates that utilizing eigenvectors corresponding to larger singular values to build the generalization map is beneficial for extracting generalized information without introducing extra parameters, thereby boosting the performance of FSOD.

**Analysis of the Number of Codewords in Dictionary Learning.** In this paper, we define a codebook containing multiple codewords to sufficiently capture category-related discriminative information from the discrimination map corresponding to relatively smaller singular values. In Table 4, we analyze the impact of using a different number of codewords. We can see that when the number is small, e.g., 16, the performance decreases. The reason may be that a small number of codewords

Table 5: The performance (%) of base and new object categories.

| Method | Base AP50 | | | New AP50 | | |
|---|---|---|---|---|---|---|
| | 1 | 3 | 5 | 1 | 3 | 5 |
| MPSR [35] | 59.9 | 68.5 | 69.4 | 40.7 | 48.9 | 53.6 |
| MPSR + Ours | 61.3 | 69.4 | 69.8 | 41.5 | **51.5** | 57.7 |
| FSCE [30] | 78.3 | 74.2 | 76.6 | 44.2 | 45.7 | 58.3 |
| FSCE + Ours | **78.6** | **74.8** | **77.8** | **46.1** | 48.9 | **60.0** |

are not sufficient to capture much discriminative information, which affects the detection performance.

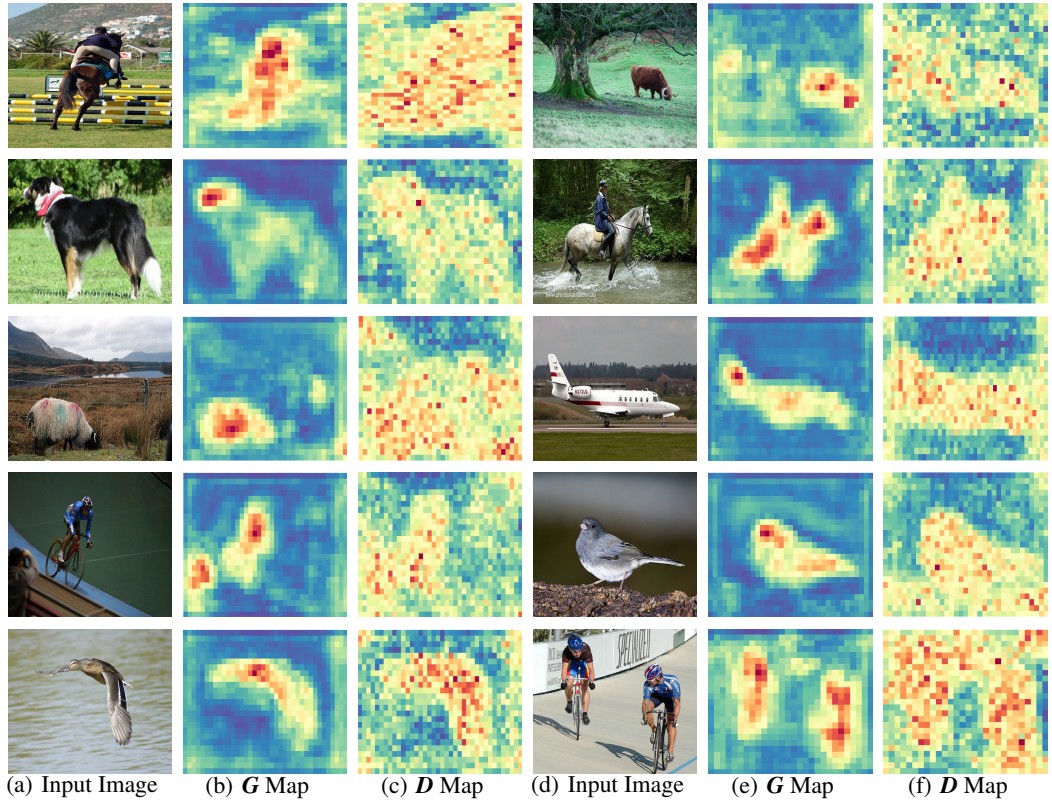

| (a) Input Image | (b) $G$ Map | (c) $D$ Map | (d) Input Image | (e) $G$ Map | (f) $D$ Map |

Figure 4: Visualization of the generalization ($G$) map and discrimination ($D$) map. The first, second, third, fourth, and fifth rows separately denote 1-shot, 2-shot, 3-shot, 5-shot, and 10-shot cases. For each feature map, the channels corresponding to the maximum value are selected for visualization.

Besides, when the number is large, e.g., 32, the performance also decreases. The reason may be that employing more codewords increases the parameters, which leads to overfitting on new object categories. For our method, the performance of using 24 codewords is the best.

**The Performance of Base Object Categories.** In Table 5, we can see that plugging our method into MPSR [35] and FSCE [30] improves not only the performance of new object categories but also the performance of base object categories. This further shows our method is beneficial for enhancing generalization and discrimination, which is conductive to the improvement of detection performance.

### 4.4 Visualization Analysis

In Fig. 4, we give visualization examples of the generalization ($G$) map and discrimination ($D$) map. We can see that the generalization map corresponding to large singular values focuses on the representative object characteristics, e.g., the head of the dog and bird, which are helpful for improving generalization and accuracy of localization. Meanwhile, the discrimination map contains rich information of background and object, which enables the following dictionary learning to sufficiently capture high-level discriminative information. This further shows that our method is effective to enhance both the generalization and discrimination abilities for FSOD.

## 5 Conclusion

In this paper, for FSOD, we focus on improving generalization and discrimination via SVD-Dictionary enhancement. Specifically, the eigenvectors corresponding to larger singular values are used to calculate a generalization map. And the eigenvectors corresponding to relatively smaller singular values are garnered to compute a discrimination map. Meanwhile, dictionary learning is employed to capture high-level discriminative information from the discrimination map. The experimental results and visualization analysis demonstrate the superiorities of our proposed method.

## Acknowledgement

Our work was supported in part by the National Natural Science Foundation of China under Grant 62132016, Grant 62171343, Grant 62071361, and Grant 62102293, in part by Key Research and Development Program of Shaanxi under Grant 2021ZDLGY01-03, and in part by the Fundamental Research Funds for the Central Universities ZDRC2102.

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
