# OpenReview forum: "Generalized and Discriminative Few-Shot Object Detection via SVD-Dictionary Enhancement"
_NeurIPS.cc/2021/Conference — NeurIPS 2021 Poster_

### Official Review · Reviewer_MNCx · 2021-07-15

**Rating:** 6
**Confidence:** 4

**Summary:**

This paper presents a solution for addressing the few-shot object detection problem through enhancing the generalization and discriminability capability, which relies on the Singular Value Decomposition (SVD). Moreover, the dictionary learning is employed to further enhance the high-level discriminative capability.

**Limitations And Societal Impact:**

N.A.

**Main Review:**

Pros:

1, This paper brings the methods for both enhancing the generalization and discriminability using SVD and dictionary learning from the research in domain adaptation, it is with novelty for adding such mechanisms into the detection structure to improve the few-shot object detection.

2, The concept of generalization (transferability) and discriminability is beneficial to few-shot object detection.

3, The experiment results is solid with obvious improvement compared to the two baseline model.

Cons:

1, The function of adding of SVD-based generalization and discriminability enhancement module is more like to improve the backbone capability of the object detection network, which is not directly related to few-shot object detection. It is also not quite clear why the RPN does not need the discrimination information for generating the proposals. Such discrimination information, e.g., the structures of objects as mentioned by the author, should be also important for generating reasonable object proposals.

2, The author mentioned the dictionary learning is beneficial for strengthening the discrimination ability of detectors. However, in my view, for few-shot object detection, the dictionary learning may more helpful to address the category imbalance in few shot since the underline principle of dictionary learning is a bit like the k-means.

3, The larger singular values contribute to the generalization ability is based on the requirement that the network feature extraction module already has some extent transferability between the source domain and the target domain. In the original SVD paper [3], such transferability is obtained by the domain adversarial training. However, in the few-shot object detection, such transferability is not certainly guaranteed, which raises the concern on the transferability between base and novel categories.

4, According the SVD spectral theory, a discriminative classifier/regressor has necessary dependence on most dimensions instead of those with only top singular value. To address this, the original SVD paper [3] added the spectral penalization to minimize the top k singular values for both source and domain to prevent it from standing out. In this paper, E in Eq. (1) is used for RPN instead of F. This raises the concern if such way will harm the discriminability for the RPN as those features belong to the top k singular values is additionally larger.


**Time Spent Reviewing:**

12

---

> ### Author Response · Authors · 2021-08-08
> **Response to Reviewer MNCx**
>
> Thanks for your recognition of our work and valuable comments.
>
> Q1: “Interpretation of the SVD-based enhancement module”.
>
> A: We agree with you that the SVD-based enhancement module improves the backbone capability of the object detection network. And the module is related to few-shot object detection (FSOD). Particularly, our fine-tuning method contains two steps for FSOD. The first step is to pre-train our method based on abundant base object categories. Then, we fine-tune the pre-trained model based on few new object categories. During the fine-tuning process, the SVD-based enhancement module is not only helpful for alleviating the information loss of base object categories but also beneficial for enhancing the representation ability of new object categories, which improves the performance of new object categories and does not suffer significant degradations for base object categories.
>
> Q2: “The reason why the RPN does not need the discrimination information”.
>
> A: We agree with you that certain discriminative information, e.g., the structures of objects, is helpful for generating object proposals. However, much discriminative information, e.g., variations of style and texture, affects the generation of object proposals, which weakens the localization ability. Here, based on the New Split 1 of PASCAL VOC and MPSR [35], we make an ablation study. From the 1-shot case to the 10-shot case, compared with our method, adding discriminative information into the RPN degrades the performance by 1.0%, 3.8%, 1.7%, 2.3%, and 1.1%.
>
> Q3: “On the interpretation of dictionary learning”.
>
> A: We agree with you that the underline principle of dictionary learning is a bit like the k-means. The codebook in dictionary learning aims to capture high-level discriminative information that is independent of categories. By fusing the proposal representations and the output of the dictionary learning (see Eq. (3)), dictionary learning strengthens the discrimination ability of object representations, which improves the classification performance.
>
> Q4: “On the guarantee the transferability between base and novel categories”.
>
> A: In this paper, the fine-tuning mechanism and dictionary learning are used to guarantee transferability between base and new categories. Concretely, we first pre-train our model based on abundant base object categories. Then, based on few samples, the fine-tuning mechanism is utilized to transfer the knowledge learned from base object categories to the new object categories. Meanwhile, during fine-tuning, our module of dictionary learning could also leverage the knowledge from both base and new categories, which is helpful for improving the discrimination ability.
>
> Q5: “Interpretation of Eq. (1)”.
>
> A: Eq. (1) does not harm the discriminability for RPN. For generating object proposals, RPN is only required to distinguish objects from background. Due to few samples, the information of new object categories in feature map F  is easily affected by base objects and background, which makes RPN could not accurately extract the proposals of new objects. To this end, taking E as the input not only strengthens the object representation ability but also retains the discriminative information in F, which is beneficial for improving detection performance.

---

### Official Review · Reviewer_2Bts · 2021-07-19

**Rating:** 5
**Confidence:** 4

**Summary:**

The paper proposes a method for few shot object detection. It splits the feature map obtained from the backbone network into two parts. First is the component with largest valued singular values (similar to low rank matrix approximation with SVD), while other is that with the remaining lower ones. It then performs dictionary learning for coding the each column of the second matrix wrt the learned dictionary and uses the code/weight matrix as features. It fuses this with the earlier matrix and passes it to the region proposal network. Further it ‘remaps’ the object proposal features the same dictionary.

**Main Review:**

The method seems complicated and perhaps the steps are not sufficiently motivated. In particular the part after l149 is confusing. P is said to be the region proposals with dimension (z,m,o,o) with z number of and o spatial size of proposals—this is not clear. Further why are these proposals coded with a dictionary which was learned with some other data from other previous layer? Seems it works in practice but I am struggling to understand the motivation and benefits of this step.

Empirical results are provided on VOC and COCO standard settings, and the method achieves improvements when added with existing methods.

I appreciate that performances on base classes is also reported which doesn’t suffer significant degradations.

For the visualizations, is the single channels with largest average value are shown, or is it a max operation on channel dim for each spatial location? Would the behavior not be expected? Like in Eigen faces etc, the large eigen value components would have sharp focus on some features, while the lower eigen value components would be spread out. Could there be a before and after visualization wrt training. Show these maps without your architecture (with plain features SVD can also be done to generate G and D) and with your architecture, does any interesting observations emerge?

I am somehow left a bit dissatisfied by the way the method is presented and explained, it seems to work, but I would like to have some hints as to why.


**Time Spent Reviewing:**

4

---

> ### Author Response · Authors · 2021-08-08
> **Response to Reviewer 2Bts**
>
> Sorry for confusing you. We will modify our paper carefully.
>
> Q1: “On the complication of our method”.
>
> A: To alleviate the impact of few samples, we explore employing SVD to boost both the generalization and discrimination abilities. Firstly, the eigenvectors corresponding to larger singular values are used to build the generalization map in which localization is performed. It is worth noting that we do not introduce extra parameters in this step. Then, the eigenvectors corresponding to relatively smaller singular values are utilized to build the discrimination map. Meanwhile, we utilize dictionary learning [40, 14] to capture high-level discrimination information from the discrimination map, which is helpful for enhancing the classification ability. To further promote the codebook in dictionary learning to retain more category-related characteristics, we try to remap the RPN output to the dictionary space and perform classification. It is worth noting that the remapping step is only used in the training stage. Since our method does not introduce complicated operations and many extra parameters, this method is easily plugged into FSOD detectors, which improves the performance.
>
> Q2: “On the RPN output P ”.
>
> A: In this paper, ‘RPN' that is a module in Faster R-CNN [27] denotes Region-Proposal Network with RoI Pooling (see the title in Fig. 2). The dimension of the output P is z×m×o×o, where z, m, and o separately denote the number of proposals, the dimension of features, and the spatial size. In experiments, z, m, and o are separately set to 1000, 256, and 8.
>
> Q3: “The reason of using dictionary to code proposals”.
>
> A: Since the RPN output P contains object-level information, using dictionary to code P can directly promote the dictionary to learn object-related information, which is beneficial for enhancing the discrimination ability.
>
> Q4: “On the visualization analysis”.
>
> A: In Fig. 4, it is a max operation on channel-dimension for each spatial location. As similar as Eigen Faces, the generalization map corresponding to large singular values focuses on the representative object characteristics, e.g., the head of the dog and bird, which are helpful for improving generalization and accuracy of localization. Meanwhile, the discrimination map corresponding to small singular values spreads out and contains rich information of background and object, which is beneficial for capturing sufficient high-level discriminative information and improves the classification ability (see Lines 306-312).
>
> Q5: “On the interesting observations of using plain features SVD”.
>
> A: Thanks for your advice. We will show more visualization examples that using plain features SVD and our method in the revision. The generalization map of using our method and that of using plain features SVD usually focus on different object regions. Taking the dog as an example, our method focuses on the head region (see the first example in the second row of Fig. 4). Using plain features SVD focuses on the body region.

---

### Official Review · Reviewer_t3Kd · 2021-08-01

**Rating:** 6
**Confidence:** 4

**Summary:**

This paper proposes an SVD-dictionary learning approach for few-shot object detection, which decomposes the input image feature into a generalization map with large eigenvalues and a discriminative map with small eigenvalues, and the decomposed feature maps are used to enhance the model generalization and discrimination.  The proposed method is evaluated on standard benchmarks using PASCAL VOC and COCO.

**Main Review:**

1. Originality: Are the tasks or methods new? Is the work a novel combination of well-known techniques? (This can be valuable!) Is it clear how this work differs from previous contributions? Is related work adequately cited?

The few-shot object detection task is a relevant new task in the field. The proposed method using SVD in FSOD is novel.  The related work is well discussed.  I like the idea of decomposing the feature space to tackle generalization and discrimination within the categories.  It would be nice to provide more motivation and empirical examples to quantify the effect. What if we only have one branch rather than both branches? It would be good to have an ablation study and a motivating example to help the readers understand.

2. Quality: Is the submission technically sound? Are claims well supported (e.g., by theoretical analysis or experimental results)? Are the methods used appropriate? Is this a complete piece of work or work in progress? Are the authors careful and honest about evaluating both the strengths and weaknesses of their work?

The submission is technically sound and provides ablation studies for the hyperparameter choices in the method. However, it would be great to show the evaluation on LVIS as it provides more categories and more realistic novel/base class splits.


3. Clarity: Is the submission clearly written? Is it well organized? (If not, please make constructive suggestions for improving its clarity.) Does it adequately inform the reader? (Note that a superbly written paper provides enough information for an expert reader to reproduce its results.)

The manuscript is well written and easy to follow.

4. Significance: Are the results important? Are others (researchers or practitioners) likely to use the ideas or build on them? Does the submission address a difficult task in a better way than previous work? Does it advance the state of the art in a demonstrable way? Does it provide unique data, unique conclusions about existing data, or a unique theoretical or experimental approach?

The proposed method outperforms previous methods.  The decomposition idea might be inspiring to others in the field.

**Time Spent Reviewing:**

5

---

> ### Author Response · Authors · 2021-08-08
> **Response to Reviewer t3Kd**
>
> Thanks for your recognition of our work and helpful comments.
>
> Q1: “On the motivation of our work”.
>
> A: Few-shot object detection (FSOD) aims to improve the detection performance for new objects that contain few annotated training samples. The few samples usually lead detectors can not well learn generalized and discriminative object-level representations, which weakens the performance of new objects. Here, the generalization is that the learned representations could alleviate the impact of certain variations (e.g., the variation of styles) [3], which is helpful for capturing representative object characteristics and improving the localization accuracy. The discrimination denotes the learned representations are discriminative, which is beneficial for enhancing the classification performance. To this end, enhancing the generalization and discrimination abilities of detectors is effective to alleviate the impact of few samples, which is beneficial for FSOD.
>
> Q2: “On empirical examples”.
>
> A: Thanks for your advice. We will provide more empirical examples in the revision. In experiments, embedding our method can improve the detection accuracy obviously. Besides, with the help of visualization, we observe that the generalization map focuses on the representative object characteristics, e.g., the body of the sheep and horse, which are helpful for improving generalization and accuracy of localization. Meanwhile, the discrimination map contains rich information of background and object, which is beneficial for enhancing the classification ability of detectors. This shows our method is indeed helpful for FSOD.
>
> Q3: “On the performance of using one branch”.
>
> A: Based on the New Split 1 of PASCAL VOC, we make an ablation study of only using the generalization branch and discrimination branch. And other components, e.g., the backbone network, are kept unchanged. From the 1-shot case to the 10-shot case, plugging the generalization branch into MPSR [35] improves MPSR’s performance by 0.6%, 5.6%, 2.3%, 3.8%, and 0.7%, which demonstrates the effectiveness of the generalization branch. Meanwhile, plugging the discrimination branch into MPSR improves MPSR’s performance by  0.5%, 3.2%, 1.2%, 1.1%, and 0.4%, which shows the effectiveness of the discrimination branch.
>
> Q4: “On the performance of LVIS dataset”.
>
> A: Thanks for your advice. Here, we plug our method into BAGS [42]. Based on LVIS dataset [43], plugging our method improves BAGS’s performance by 0.8% in terms of mAP metric, which demonstrates that enhancing generalization and discrimination is effective for long-tail object detection task.
>
> [42] Li Y, Wang T, Kang B, et al. Overcoming classifier imbalance for long-tail object detection with balanced group softmax. Proceedings of the IEEE/CVF conference on computer vision and pattern recognition. 2020: 10991-11000.
>
> [43] Gupta A, Dollar P, Girshick R. LVIS: A dataset for large vocabulary instance segmentation. Proceedings of the IEEE/CVF Conference on Computer Vision and Pattern Recognition. 2019: 5356-5364.

---

### Decision · Program_Chairs · 2021-09-27

**Decision:**

Accept (Poster)

**Comment:**

This paper offers an SVD based decomposition of a learned representation, factoring an embedding into components that are optimal for localization vs description.  On balance the reviewers find the paper acceptable, with a majority of reviewers including the most knowledgeable reviewer in the area supporting acceptance, due to the novelty of the method and the improved performance.  The AC concurs with these reviewers.